# Effect of malaria and HIV/AIDS co-infection on red blood cell indices and its relation with the CD4 level of patients on HAART in Bench Sheko Zone, Southwest Ethiopia

Solomon Ejigu[1]*, Diresbachew Haile[2], Yerukneh Solomon[3]

1 Department of Biomedical Sciences, Collage of Medicine and Health science, Mizan-Tepi University, Mizan-Aman, Ethiopia, 2 Department of Physiology, School of Medicine, Collage of Health Science, Addis Ababa University, Addis Ababa, Ethiopia, 3 Department of Biomedical Sciences, Collage of Medicine and Health Science, Debre Berhan University, Debre Berhan, Ethiopia

* solomonejigu67@gmail.com

## Abstract

### Background

Malaria and HIV/AIDS are the two most common infections in sub Saharan Africa (SSA) and worldwide. HIV infected individuals in malaria endemic areas experience severe malaria episodes. The immunological basis of this clinical observation is unclear and the hematologic abnormalities such as anemia in malaria and HIV co infected patients were inconsistent from studies in the past. Ethiopia's three-fourth of the landmass is malarious and HIV prevalence is high that significantly affect RBC indices and other hematologic profiles.

### Objective

This study aimed to compare RBC indices and anemia in HIV patients' co-infected with malaria and those HIV patients without malaria and correlates these with CD4 level.

### Methods

A comparative cross-sectional study was employed on 103 malaria-HIV/AIDS co infected (MHC) and 103 HIV patients without malaria on HAART of the same ART centers in Bench Sheko Zone. Data was collected by structured questionnaire and blood samples were collected from both groups for malaria test and RBC indices measurement. Data was entered and checked in Epi-data and exported to IBM SPSS version 21 software packages for analysis.

### Results

There were significant differences in Mean±SD of RBC indices between the two groups (P<0.001). RBC, Hgb, HCT and MCV were lower in MHC patients. In total study participants, significant positive correlation was observed between CD4 count with MCV, CD4 count with MCH and CD4 count with anemia. In the group of malaria-HIV co-infected, CD4

**Data Availability Statement:** Data within the paper and its Supporting Information files.

**Funding:** No fund received.

**Competing interests:** No competing of interest.

count with RBC and CD4 count with Hgb and in HIV without malaria CD4 count with MCV, CD4 count with MCH and CD4 count with MCHC were positively correlated. Overall anemia prevalence was 45.1%. Anemia prevalence in MHC (Malaria-HIV co-infected) was 63.4%. Anemia prevalence distribution among sex showed that 61.3% in female sex and anemia prevalence distribution among CD4 group showed 55.9% in patients with CD4 count of ≤500 cells/µl. Anemia in MHC patients was higher in those with CD4 count of ≤500 cells/µl (59.3%) while in OH (Only HIV infected) anemia prevalence was similar in those with CD4 count of ≤500 and ≥500 cells/µl (50%). There is significant difference in anemia in MHC and OH infected with different CD4 group (P<0.01).

## Conclusion

There was a difference in RBC indices in both groups; RBC, Hgb, HCT and MCV were lower in MHC patients. There was positive correlation between CD4 counts with some RBC indices in combined both groups. However, there was positive correlation between CD4 counts with RBC and Hgb in malaria-HIV co-infected. The combined prevalence of anemia was higher and anemia in MHC was greater than OH infected patients.

## Introduction

Malaria and HIV/AIDS are the two most common infections in sub-Saharan Africa (SSA). They overlap globally because a number of HIV-infected individuals live in regions with malaria transmission, therefore, defining their prevalence and outcome will be important [1]. Some studies suggest the synergistic and bi-directional interaction of both HIV and malaria infection [2]. Both kill millions of people yearly with a heavy burden on Africa, India, Southeast Asia and South America [3]. HIV reduces the host's immunity to infecting agents of malaria, this expands both diseases in areas where their burden is high [4] and their interaction will have profound public health effect [5]. Ethiopia's three-fourth of the land mass is endemic for malaria, mainly *Plasmodium falciparum* and *Plasmodium vivax* [6]. Although malaria and HIV are diseases of mortality that affects people in Ethiopia [7], their burden is not well documented [8].

Malaria, which is transmitted by the bite of female *anopheles* mosquito [9], is a life–threatening vector-borne *plasmodium* parasitic tropical infectious disease [10, 11]. Malaria results in 300–500 million cases and 1.5–2.7 million deaths annually [12], of which 80% is from Africa [13]. In Ethiopia, about 68% of the total population is living in malaria risky area which makes it to be top ranking disease in Ethiopia [14, 15]. Hematological changes are the most common complications in malaria [12]. Anemia is another complication of malaria [16]. HIV is the etiologic agent of AIDS and AIDS-related Complexes (ARC) [17]. In HIV/AIDS comorbidity such as malaria are usual and infection of additional pathogens accelerates disease progression that enhances morbidity [18]. SSA accounts >70% of total HIV infection globally [19]. In Ethiopia, newly diagnosed HIV/AIDS patients in 2016 was 30,000 and people living with HIV/AIDS was 710,000 [20]. HIV depletes $CD^4$+T lymphocytes that increases risk of opportunistic infections [7]. Malaria and HIV infections are the two disease conditions of major public health problems in many parts of the world [21, 22]. In Africa, HIV-associated severe malaria in individuals with HIV is emerging [23, 24]. In Ethiopia, people residing in regions where both diseases are highly endemic are prone to develop co-infection [16, 25]. However, there is a lack of

clarity on physiologic impact of both infections [26]. Evidences indicated that reduced immunity due to the virus causes clinical attacks of malaria [16] which leads to anemia [27]. Anemia was documented as an independent predictor of morbidity and mortality in malaria-HIV/AIDS co infection and their effect is severe in immunosuppressed patients [28]. Studies on MHC association with anemia are inconsistent.

Some studies reported the negative effect of dual infection on hemoglobin [29] whereas others reported differently [30, 31]. There is limited information on the collective impact of MHC on the hemoglobin levels and other RBC profiles [22]. In Ethiopia, studies on MHC are very limited and no study of this type in this area although both infections are the most common public health problems [32]. This study aimed to compare RBC indices, anemia, and correlation of RBC indices and CD4 count in malaria-HIV co-infection as compared to HIV without malaria patients who were on HAART in Bench Sheko Zone.

## Materials and methods

### Study area

This study was conducted at ART units of health institutions in Bench Sheko Zone, Southwest Ethiopia.

### Source and study populations

The source populations were all HIV/AIDS patients with regular follow up on HAART while the study populations were HIV/AIDS patients who visited the health facilities during the study period.

### Sample size determination

The study was conducted on 103 MHC and 103 only HIV infected patients with the total sample size of 206. The sample size for this study was determined based on the formula for the double population proportion.

The following two-sided population proportion formula was used for the determination of the sample size for this particular study.

$$n = \frac{(p^-)\ (1-p^-)\ (Z\alpha\ /2 + Z\beta)^2 r + 1/r}{E^2} \qquad Where, p^- = (p1 + p2)/2.$$

$p^-$ (1- $p^-$) is a measure of variability, E = p1-p2 is the effect size that is the difference in means or proportions, n = required minimum sample size, P1 (Prevalence of event in cases) = 0.60, P2 (Prevalence of event in controls) = 0.40, $E^2$ (effect size difference in proportion) = $(0.6–0.4)^2 = 0.04$

$p^- = (P1 + P2)/2 = (0.6 + 0.4)/2 = 0.5$ and $1 - p^-_{=1-0.5=0.5}$, $Z_\alpha$ (at 95% level of significance) = 1.96

$Z_\beta$ (for 80% power of the test) $= 0.84$, $r$ is proportion of controls to cases $= 1$

This figures were taken from previous study entitled "Effect of malaria infection on hematological profiles of people living with human immunodeficiency virus(HIV) in Gambella, Southwest Ethiopia" [28]. Therefore, based on this assumption the sample size was calculated

as follows.

$$\mathbf{n} = \frac{\mathbf{(0.5) \times (0.5) \times (1.96 + 0.84)2}}{\mathbf{(0.6 - 0.4)2}}[1 + 1]/1$$

$$\mathrm{n} = \frac{(0.25) \times (7.8) \times (2)}{0.04} \simeq 98$$

Considering double proportion, the required sample obtained will become **196** study participants and including a 5% non-response rate the total sample size was made to be **206**. Among them **103** participants were Malaria-HIV/AIDS co-infected (MHC) while the remaining **103** were HIV/AIDS patients without co-infection.

## Methods of data collection

A structured questionnaire, Ethylene DiameTetracaetic Acid tube (EDTA), Microscope, alcohol swab, syringes, surgical Glove, Ice bag and Hematology analyzer were used during the study. The inclusion criteria for HIV/AIDS patients were those who were scheduled to visit the ART units for routine medical review. For malaria infected HIV patients, clinical manifestations of malaria such as febrile illness and patients who were not using anti-malarial drugs for past 2 months. Patients with no risk for chronic diseases and consented participants were included in both groups.

## Ethical clearance

Prior to data collection ethical clearance was obtained from Research and Ethics Committee of Medical Physiology department, Addis Ababa University. Subjects were informed about the aim, procedures and side effects in relation to minor-invasive but not harmful procedures. Following the orientation and further feedback to their doubt, written and oral informed consent were secured with the option to withdraw from participation in the study without any precondition at any time.

## Study design and period

A comparative cross- sectional study design was employed from April to June, 2019. Three health institutions (Biftu health center, Sheko health center and Mizan-Tepi University Teaching Hospital).

## Sampling technique

Systemic random sampling technique was employed to select only-HIV infected (OH) patients and incidental sampling technique was used to recruit malaria-HIV co-infected (MHC). The data were collected by trained health professionals using standardized questionnaire under the supervision of the principal investigator. Under aseptic condition, 5ml of blood was drawn for Microscopic malaria test and the remaining blood was kept into EDTA tube at temperature of 8-10˚c for complete blood count (CBC). For securing the quality of the laboratory data the standard operating procedure (SOP) for each test was followed. For CBC measurement Automated Hematology analyzer was used with strict adherence to the manufacturer instruction.

## Data entry and analysis

Data was entered in to Epi-Data 3.1 and exported into SPSS version 21 for analysis of the data.

Independent samples T- test, Chi square test, Pearson correlation were used to analyze the data and p< 0.05 was taken as significance level.

## Results

Totally 206 study participants completed the study, those who were consented to be interviewed and gave their blood sample. The average age of the participants was 35.33±7.8. Female participants were 53.9% in total study participants and their percent were higher in both groups as compared to male counterparts, 53.4% were married and 11.1% were single. Rural and urban dwellers were 37.8% and 62.2% respectively. Among rural dwellers 26.0% were MHC and out of 62.2% urban settlers 38.4% were OH.

*See* Table 2: A total of 71.3% used ART for ≥3 years and 28.7% used ART for ≤ 2 years. In MHC 69% used ART for ≥3 years while 73.8% of OH used ART for ≥ 3 years (P<0.02, 95% CI). Totally 54.8% had CD4 count of ≥500 cells/μl (22.8% MHC vs. 32% OH), 35.4% were with CD4 count between 200–499 cells/μl, of which 24.7% were MHC and patients with CD4 count of ≤ 200 cells/μl were 9.7% with majority (15/20) being OH infected. From only HIV infected, majority (64.07%) had CD4 count of ≥500 cells/μl as compared with 45.6% of MHC of similar CD4 count. MHC patients with CD4 count between 200–499 cells/μl were greater than that of OH (49.5% vs. 21.4%) and there is significant difference in CD4 count of MHC and HIV alone (P<0 .01) (Tables 1 and 2).

There was a significant difference in RBC indices of MHC and OH infected (P≤ 0.001, 95% CI). The mean ± SD of RBC count, Hgb and HCT of MHC and OH was different significantly (P≤ 0.01). Similarly MCV, MCH, MCHC and RDW in MHC and OH infected differ significantly (P≤ 0.01, 95% CI). RBC, Hgb, HCT and MCV were lower in MHC relative to OH infected (P≤0.01) (Table 3). *Abbreviations are indicated under each table below.*

There was no significant correlation between CD4 count with RBC, Hgb and MCHC. However, there was significant positive correlation between CD4 count and anemia [r = 0.16, P = 0.02, 95% CI]. In MHC there was significant positive correlation between CD4 count with RBC and Hgb only, RBC [r = 0.29, P = 0.03, 95% CI] and Hgb [r = 0.19, P = 0.048, 95% CI]. In OH infected patients positive correlation was observed between CD4 count with MCV, MCH and MCHC (Table 4). Abbreviations are indicated under each table below.

Anemia is defined as Hgb level of ≤12g/dl for females and ≤13g/dl for adult males based on WHO anemia classification [33]. The combined prevalence of anemia was 45.1%. Anemia in MHC was 63.4% while 36.6% in OH among anemic. Anemia was higher in MHC as compared to OH infected (P = 0.01). From combined prevalence of anemia in both groups; anemia prevalence in female was higher than in male (61.3% vs.38.7%) among 45.1% anemic totally. Among anemic female, 71.9% were MHC and 28.07% were OH. However, in male anemia was similar in MHC and OH (50 %) (Table 5).

### Association between anemia with ART Regimen and CD4 count

Anemia prevalence in 1st line ART drugs users was 80.6% while 19.4% in 2nd line ART drug users. Anemia prevalence in patients with CD4 count of 200–499 cells/μl was 45.1% (32.2% in MHC and 12.9% in OH) and it varies significantly in CD4 count between 200–499 in MHC and OH (P≤0.01, 99% CI). Anemia was lower in patients with CD4 count ≤200, 10.7% (5.4% in both groups). Anemia prevalence differs in MHC and OH infected in patients with CD4 count ≤200 (P≤0.03). In patients with CD4 count ≥500 anemia was 44.08% (25.8% in MHC and 18.3% in OH) but anemia was 55.9% in those with CD4 count of ≤500. The prevalence of anemia in MHC patients was higher in CD4 count of ≤500 cells/μl (59.3%) as compared to those with ≥500 (40.7%). However, in OH infected, anemia in patients with the CD4 counts of

**Table 1. Socio-demographic characteristics of the study participants, Bench Maji Zone 2019.**

| Variables | MHC | OH | Total (%) |
|---|---|---|---|
| | (N = 103) | (N = 103) | (N = 206) |
| Age (In years) | | | |
| 10–20 | 4(3.9%) | 4(3.9%) | 8(3.9%) |
| 21–30 | 34 (33%) | 24 (23.3%) | 58(28.2%) |
| 31–40 | 39 (37.9%) | 46 (44.7%) | 85(41.3%) |
| 41–50 | 26(25.2%) | 29 (28.2%) | 55(26.7%) |
| Sex | | | |
| Male | 45 (43.7%) | 50 (48.5%) | 95(46.1%) |
| Female | 58(56.3%) | 53 (51.5%) | 111(53.9%) |
| Marital status | | | |
| Married | 63(61.16%) | 47 (45.63%) | 110(53.4%)** |
| Single | 18 (17.47%) | 5(4.85%) | 23(11.1%) |
| Divorced | 18(17.47%) | 37(35.9%) | 55(26.7%) |
| Widowed | 4(3.88%) | 14(13.6%) | 18(8.73%) |
| Occupation | | | |
| Government employee | 26(25.24%) | 15(14.56%) | 41(20%) |
| Self employed | 12(11.65%) | 21(20.4%) | 33(16%) |
| Merchant | 30(29.12%) | 36(34.95%) | 66(32.04%) |
| Farmer | 25(24.3%) | 26(25.24%) | 51(24.76%) |
| Other | 10(9.7%) | 5(4.9%) | 15(7.28%) |
| Residency | | | |
| Rural | 54(52.4%) | 24(23.3%) | 78(37.9%) * |
| Urban | 49(47.6%) | 79(76.7%) | 128(62.1%) |
| Educational Status | | | |
| Uneducated | 13(12.62%) | 25(24.27%) | 38(18.5%) * |
| Primary school | 46(44.66%) | 44(42.72%) | 90(43.7%) |
| Secondary school | 15(14.56%) | 19(18.45%) | 34(16.5%) |
| Above secondary | 29(28.16%) | 15(14.56%) | 44(21.4%) |
| Malaria prevention Method | | | |
| Anti-malarial tablets | 15(14.56%) | 6(5.82%) | 21(10.19%) * |
| Bed nets | 74(71.84%) | 77(74.75%) | 151(73.3%) |
| Environmental sanitation | 6(5.82%) | 16(15.53%) | 22(10.68%) |
| Other means | 8(7.77%) | 4(3.88%) | 12(5.83%) |
| Bed-net use per week | | | |
| Once per week | 10(11.62%) | 5(5.61%) | 15(8.57%) * |
| Twice per week | 20(23.25%) | 8(8.98%) | 28(16%) |
| Three times per week | 14(16.3%) | 3(3.4%) | 17(9.7%) |
| ≥ Four times per week | 42(48.8%) | 73(82%) | 115(65.7%) |

≤500 cells/μl and ≥500 cells/μl were each 50% (17/34). There were significant difference in anemia in MHC and OH infected with different CD4 group (P≤0.01, 99% CI).

## Characteristics of anemia in MHC and OH infected patients

Type of anemia was defined based on MCV and MCHC. Microcytosis (MCV< 80 fl), macro-cytosis (MCV >100 fl), normocytic (MCV 80-100fl) and hypochromic was defined as MCHC value < 31 g/dl [34]. From anemic, 34.4% were macrocytic [19.4% MHC vs.15% of OH] while 65.6% were normocytic [44.1% MHC vs. 21.5% of OH] and no microcytic type anemia. The

**Table 2. Clinical characteristics of the participants, Bench Sheko Zone 2019.**

| Variables | MHC | OH | Total | P-value |
|---|---|---|---|---|
| | (N = 103) | (N = 103) | (N = 206) | |
| Duration on ART | | | | |
| For 1 years | 12(11.6%) | 8(7.7%) | 20(9.7%) | 0.027* |
| For 2 years | 20(19.4%) | 19(18.5%) | 39(18.9%) | |
| For 3 years | 16(15.5%) | 8(7.7%) | 24(11.6%) | |
| For 4 years | 30(29.1%) | 22(21.4%) | 52(25.2%) | |
| ≥5 years | 25(24.3%) | 46(44.7%) | 71(34.5%) | |
| ART Regimen | | | | |
| 1st Line ART drugs | 81(78.6%) | 89(86.4%) | 170(82.5%) | 0.2 |
| 2nd Line ART drugs | 22(21.4%) | 14(13.6%) | 36(17.5%) | |
| HIV/AIDS Stage | | | | |
| Stage I | 97(94.2%) | 96(93.2%) | 193(93.7%) | 0.5 |
| Stage II | 5(4.8%) | 4(3.9%) | 9(4.37%) | |
| Stage III | 1(0.97%) | 3(2.9%) | 4(1.94%) | |
| CD4 Count | | | | |
| ≤200 cells/μl | 5(4.85%) | 15(14.56%) | 20(9.7%) | 0.000** |
| 200–499 cells/μl | 51(49.51%) | 22(21.36%) | 73(35.44%) | |
| ≥500 cells/μl | 47(45.63%) | 66(64.07%) | 113(54.8%) | |

Results are expressed both in total number (N) and percentages (%), MHC = Malaria-HIV/AIDS co infected, OH = Only HIV infected,

* = The variable significantly differs in MHC and OH with significance level of P≤0.05,

** = P≤0.001.

anemia type in both groups varies significantly (P<0.01). Hypochromic anemia was 82.8% [52.7% MHC vs. 30.1% OH] and normochromic were 17.2% [10.8% MHC vs. 6.4% OH]. Of 45.1% of anemia totally, normocytic normochromic, normocytic hypochromic, macrocytic hypochromic and macrocytic normochromic anemia was 16.1%, 49.5%, 33.3% and 1% respectively.

## Discussion

Malaria-HIV/AIDS co-infection (MHC) is found to have a greater effect in reducing RBC indices such as Hgb and HCT compared to single infection of either type due to the effect of both pathogens [35]. We found a significant difference in mean value of RBC indices in MHC and OH infected patients (P = 0.01). Similar to our study, Nigerian study showed significant differences in RBC indices in MHC and OH [4], Tchinda; et al., [35] also showed RBC and Hgb to be different in MHC and OH infected patients. Another Nigerian study reported similarly regarding Hgb in both group (P<0.05, 95% CI) [36]. A study from Southwest Ethiopia, Gambella reported a significant differences in mean ±SD of Hgb and HCT in MHC and OH (P≤0.02) [28]. The mean ± SD of RBC, Hgb, HCT and MCV were lower significantly in MHC relative to OH. This agrees with the study of Nigeria [4]. Study of Ghana showed lower Hgb in MHC than OH [37]. In Ethiopia, study reports lower Hgb and HCT in MHC than in OH [28]. Ethiopian study also showed decreased Hgb and HCT in MHC (P≤0.04, 95% CI) [32]. There was no significant correlation in CD4 count with RBC while positive correlation with Hgb, MCHC, MCV, MCH and anemia status in total study participants. Study from USA [38], Iran [39], Ghana [40] and Uganda [41] reported an association between lower CD4 count and anemia. Study from Cameroon [42] and Ethiopia demonstrated correlation of CD4 count and

**Table 3. RBC indices of MHC as compared with OH infected patients, Bench Sheko Zone 2019.**

| RBC Indices | MHC (N = 103) | OH (N = 103) | Mean | P-value |
|---|---|---|---|---|
| | Mean±SD | Mean±SD | Difference | |
| RBC (×10 $^6$/μl) | 3.93±0.48 | 4.35±0.88 | -0.42 | P≤0.001 |
| Hgb (g/dl) | 12.14±1.90 | 13.70±3.83 | -1.56 | P≤0.001 |
| HCT (%) | 39.44±5.54 | 44.57±9.36 | -5.13 | P≤0.001 |
| MCV (fl) | 101.13±10.64 | 103.72±14.86 | -2.59 | P≤0.001 |
| MCH (pg) | 31.19±3.75 | 31.05±4.02 | +0.14 | P≤0.001 |
| MCHC (g/dl) | 30.59±2.13 | 29.91± 1.62 | +0.68 | P≤0.001 |
| RDW (%) | 13.52±1.30 | 13.49± 1.17 | +0.03 | P≤0.001 |

Results are expressed in Mean ± SD, SD = Standard Deviation, [**RBC** = Red Blood Cell, **HCT** = Hematocrit, **MCH** = Mean Cell Hemoglobin, **RDW** = Red cell distribution Width, **Hgb** = Hemoglobin, **MCV** = Mean Cell Volume, **MCHC** = Mean Cell Hemoglobin Concentration].

anemia in HIV infected patients before starting ART [43]. But studies of Eastern India and Northwest Nigeria didn't find any association [44–46]. This difference might be due to variation in medication type used in HIV patients, socio demographics and nutritional status.

The positive correlation between CD4 count and anemia might be due to leukemia in HIV patients where an increased production of lymphocytes increases production of WBC that could increase CD4 and this suppresses RBC to cause anemia. In MHC there was significant positive correlation between CD4 count with RBC, Hgb and HCT while in patients of OH there was positive correlation between CD4 with MCV, MCH and MCHC. And no significant correlation between CD4 count with Hgb in OH infected unlike the available studies that reported significant positive correlation between CD4 count and Hgb (r = 0.17, p = 0.01) [40]. This disagreement may be originated from the design of the study where Ghanaian study was prospective case control study and blood samples were taken from the subjects before the initiation of ART. The positive correlation between CD4 counts with RBC, Hgb and HCT among MHC could be explained most likely by the fact that decreased CD4 count as HIV disease progresses could cause anemia by myelosuppression from HIV that impairs erythropoietin [47, 48], malaria may also induce removal of parasitized RBC [49].

The combined prevalence of anemia in both groups was 45.1%. This in lines with the finding of Uganda (47.8%) [50], Nigeria (45.26%) [51], Northern Ethiopia (43%) [47] in both

**Table 4. Correlation of CD4 count with RBC indices and anemia.**

| RBC Indices | RBC | Hgb | MCV | MCH | MCHC | Anemia |
|---|---|---|---|---|---|---|
| | | | **(Both MHC and OH)** | | | |
| P. corr | -0.11 | 0.006 | 0.14 | 0.169 | 0.071 | 0.16 |
| P-value | 0.1 | 0.93 | 0.04 | 0.01 | 0.31 | 0.02 |
| | | | **MHC** | | | |
| P. corr | 0.29 | 0.19 | -0.032 | -0.07 | -0.1 | |
| P-value | 0.003** | 0.048* | 0.75 | 0.47 | 0.317 | |
| | | | **OH** | | | |
| P. corr | -0.325 | -0.038 | 0.228 | 0.350 | 0.28 | |
| P-value | 0.001* | 0.700 | 0.021 | 0.000* | 0.004* | |

*Correlation between CD4 count and RBC Indices along with anemia.

*P. corr = Pearson correlation Coefficient.

**Table 5. The prevalence of anemia among MHC and OH, Bench Maji Zone 2019.**

| Variables | | MHC | OH | Total | P-value |
|---|---|---|---|---|---|
| Anemia status | Anemic | 59(57.3%) | 34(33%) | 93 (45.14%) | 0.000** |
| | Non-anemic | 44 (42.7%) | 69 (67%) | 113 (54.85%) | |
| Anemic Male | | 18 | 18 | 36 (38.7%) | |
| Anemic Female | | 41 | 16 | 57 (61.3%) | |

groups. But anemia in this study was lower than the studies of Cameroon (56.9%), Ghana (67%) [22, 42], North east Nigeria (49.5%), Southwest Cameroon (49.6%) and China (51.9%) [52–55]. The increased anemia prevalence in above studies could be due to high proportion of female in Ghana (73%) and Cameroon (73.9%) relative to current study (53.8%) and in study of Cameroon, the patients with CD4 count of ≤200 cells/μl were higher. Furthermore, geographic, socio-demographic and cultural variation regarding nutrition in the study of China and the current study and variation in age and inclusion of non-ART subjects in the study of South west Cameroon could have created this discrepancy. In MHC anemia was 63.4% and 36.6% in OH infected. Collaborating to this Nigerian study showed (66.7% in MHC vs.33% in OH) [36], Gambella region in Ethiopia showed (58.4% in MHC vs. 41.6% in OH) [28]. Higher anemia prevalence in MHC could be due to the impact of two infections acting individually where HIV results myelosuppression while malaria causes hemolysis of RBC [35]. Anemia in female was 61.3% and 38.7% in male. From anemic female, 72% were MHC. This collaborates with Ethiopian study where 62% of women and 38% of men living with HIV/AIDS were anemic [56, 57].

Furthermore, in Northern Ethiopia 59.03% of anemia in female and 40.9% in male [47]. However, study in Ghana found unacceptably high anemia among female, 94.4% [22]. This could be due to increased HIV in the study area (3%) [58] so does malaria in Ghana [59], female participants in Ghana was 73%, factors such as socio demographic, age and nutrition related factors might pose this difference in anemia prevalence in female between this study and Ghana.

High prevalence of anemia in female could be attributed to physiologic reasons like menstrual blood loss and the drains on iron stores that occur with pregnancy and delivery [60]. In patients with CD4 count ≤200cells/μl anemia were 10.8% that was not similar with 20% from Southwestern Ethiopia [28]. This could be due to incomparable number of patients with CD4 count ≤200. Anemia in subjects with CD4 of 200–499 was 45.1%, this was higher than 39.3% of similar study of Ethiopia [28] but similar with 44% of anemia in similar CD4 count in Central Tanzania [61]. In patients with CD4 count of ≥500 anemia was 44%. This agrees with 40.5% anemia in Gambella region, Southwest Ethiopia [28]. Generally, anemia in patients with CD4 counts of ≤500 and ≥500 cells/μl was 56% and 44% respectively. This deviates from the finding of Tanzanian, 79.4% vs.20.4% and Brazilian, 61.1% vs. 29.4% [61, 62]. Elevated anemia in patients with CD4 counts of ≤500 could be due to destruction and diminished production of RBC resulting in low Hgb as HIV advances to AIDS [48]. In MHC patients, anemia was higher in CD4 count of ≤500 while there was no difference in OH infected patients having CD4 count of ≤500 and ≥500. Among MHC, we found 54.4% had CD4 count of ≤500 and 45.6% had CD4 count ≥500 while majority of OH infected were having CD4 count of ≥500 cells/μl (64.07%) (P≤0.025). Similar to this, study of Ghana reported only one patient with MHC had CD4 cell count ≥500 and the rest of MHC had CD4 cell count between 3 -512cells/ μl, Nigerian study showed that 63.8% MHC had CD4 count of ≤500. In another Nigerian study 54% patients of MHC had CD4 count of ≤500 [22, 28, 63, 64]. The incidence and effect

of malaria in HIV patients gets tremendous when the immune gets compromised as evidenced from lower CD4 count. This requires early detection of co infection and identification of associated factor for better management of co infected patients to reduce their burden like anemia as it is one of the abnormality related to both infection [47].

## Conclusion

This study showed that there was significant difference in RBC indices in MHC and OH infected patients and the mean ± SD of RBC, Hgb, HCT and MCV were lower in MHC. In MHC there was significant positive correlation between CD4 count with RBC, Hg and HCT. Anemia prevalence was undeniably higher and disproportionately higher in MHC, in female sex, in patients with CD4 count of ≤500 cells/μl. Generally, malaria had exerted a magnificent negative effect on hematological parameters of HIV patients even though they were on HAART. This study was limited due to its cross-sectional study design because it doesn't assess cause and effect relationships as the longitudinal study design does. Second, it would have been better if we would have used PCR for malaria species identification.

## Supporting information

**S1 Appendix.**
(DOCX)

**S1 Data.**
(SAV)

**S2 Data.**
(SAV)

**S3 Data.**
(SAV)

**S4 Data.**
(SAV)

**S5 Data.**
(SAV)

**S6 Data.**
(SAV)

## Acknowledgments

The authors acknowledge the Addis Ababa University and Bench Sheko Zone that permitted us to conduct the study and the participants who consented to participate in the study.

## Author Contributions

**Conceptualization:** Solomon Ejigu.

**Data curation:** Solomon Ejigu.

**Formal analysis:** Solomon Ejigu, Yerukneh Solomon.

**Funding acquisition:** Solomon Ejigu.

**Investigation:** Solomon Ejigu, Diresbachew Haile.

**Methodology:** Solomon Ejigu, Diresbachew Haile, Yerukneh Solomon.

**Resources:** Solomon Ejigu, Yerukneh Solomon.

**Software:** Solomon Ejigu, Diresbachew Haile.

**Supervision:** Diresbachew Haile, Yerukneh Solomon.

**Visualization:** Diresbachew Haile, Yerukneh Solomon.

**Writing – original draft:** Solomon Ejigu.

**Writing – review & editing:** Solomon Ejigu, Yerukneh Solomon.

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
