## [Decision Letter · Decision Letter 0]

19 Mar 2021

PONE-D-21-03985

Effect of Malaria and HIV/AIDS co-infection on Red Blood Cell Indices and Its relation with the CD4 level of Patients on HAART in Bench Sheko Zone, Southwest Ethiopia.

PLOS ONE

Dear Dr. Solomon Ejigu

Thank you for submitting your manuscript to PLOS ONE. After careful review with two experts in the fields, we feel that your manuscripts but does not meet PLOS ONE’s publication criteria as it currently stands. You are welcome to submit a revised version of the manuscript for our consideration if you could rigorously address the all points raised during the review process. However, please aware a revision does not always lead to a publication.

Please address point by point for all reviewers' comments including expert reviewer#1's detailed points as well as expert reviewer#2's general points, which I agreed. In addition, 1) please use less abbreviations or define it clearly at the beginning; 2) please spell out the gender specific Anemia criteria used;  3) please address whether there is a potential inclusion bias between the groups as we did see obviously the coinfected group seems sicker as reflected by their CD4; and 4) please acknowledge the small number of population studied.

We look forward to receiving your revised manuscript.

Kind regards,

Weijing He, M.D.

Academic Editor

PLOS ONE

Journal Requirements:

2. In the Methods, please clarify that participants provided oral consent. Please also state in the Methods:

- Why written consent could not be obtained

- Whether the Institutional Review Board (IRB) approved use of oral consent

- How oral consent was documented

For more information, please see our guidelines for human subjects research: https://journals.plos.org/plosone/s/submission-guidelines#loc-human-subjects-research

3. Please provide a sample size and power calculation in the Methods, or discuss the reasons for not performing one before study initiation.

4. Please ensure you have discussed any potential limitations of your study in the Discussion, including study design, sample size and/or potential confounders.

5. Please include additional information regarding the survey or questionnaire used in the study and ensure that you have provided sufficient details that others could replicate the analyses. For instance, if you developed a questionnaire as part of this study and it is not under a copyright more restrictive than CC-BY, please include a copy, in both the original language and English, as Supporting Information.

7. We note that you have indicated that data from this study are available upon request. PLOS only allows data to be available upon request if there are legal or ethical restrictions on sharing data publicly. For information on unacceptable data access restrictions, please see http://journals.plos.org/plosone/s/data-availability#loc-unacceptable-data-access-restrictions.

8. Your ethics statement should only appear in the Methods section of your manuscript. If your ethics statement is written in any section besides the Methods, please move it to the Methods section and delete it from any other section. Please ensure that your ethics statement is included in your manuscript, as the ethics statement entered into the online submission form will not be published alongside your manuscript.

Reviewers' comments:

Reviewer's Responses to Questions

**Comments to the Author**

1. Is the manuscript technically sound, and do the data support the conclusions?

Reviewer #1: Yes

Reviewer #2: Partly

2. Has the statistical analysis been performed appropriately and rigorously? 

Reviewer #1: Yes

Reviewer #2: Yes

3. Have the authors made all data underlying the findings in their manuscript fully available?

Reviewer #1: No

Reviewer #2: Yes

4. Is the manuscript presented in an intelligible fashion and written in standard English?

Reviewer #1: No

Reviewer #2: No

5. Review Comments to the Author

Reviewer #1: The manuscript titled "Effect of Malaria and HIV/AIDS co-infection on Red Blood Cell Indices and Its relationwith the CD4 level of Patients on HAART in Bench Sheko Zone, Southwest Ethiopia" by Solomon Ejigu et al explored Malaria and HIV co-infection and the effects on RBC of infected patients. This is a valid research. However, there is need for major corrections before manuscript can be accepted for publication. The manuscript lacks line numbering, hence it is difficult to make reference to sections of the document while reviewing it. The references are poorly done. I suggest that authors rewrite the entire references following the Journal's guidelines for references. The methods and results were also poorly presented and the discussion was not sequentially done. Please see other specific comments below:

Please correct the spelling of "co infected" to "co-infected" all through the manuscript. Do the same for co-infection.

Abstract:

Pg 2, line 5: "... HIV co infected patients were inconsistent." is this an inference made from the present study or a general setting in the study location? Please rephrase statement.

Pg 2, line 8: "in HIV patients’ co infected" should be "in HIV patients co-infected"

line 9: "and correlate these"

line 15: P ≤ 0.001? Please check and correct this. Use > or < to represent the significant difference and = for exact p value all through the manuscript.

line 16: "between CD4 count with MCV," please use appropriate grammar.

lines 16-18: "While in  MHC,  CD4  count  with  RBC  and  Hgb  and  in  HIV  without  malaria  CD4  count  with  MCV, MCH  and  MCHC  were  positively  correlated." This statement is not clear. Please recast for better understanding.

line 21: what is OH? Please write in full.

The entire result were not properly presented by the authors in the abstract. I suggest that authors use short, simple and clear sentences to describe the results for easy readability. For example, lines 19-21, authors wrote "in MHC was 63.4%, 61.3%  in female sex and 55.9% in patients with  CD4 count of ≤500 cells/μl." and went on to write "Anemia in MHC patients was  higher in those with CD4 count of ≤500  cells/μl (59.3%)" why are there two different results for the same parameter for MHC? (55.9% and 59.3%). Please revisit the result section of the abstract and represent findings in clear words.

line 26:" some  RBC  indices  in  total" please replace "in total" with a better phrase.

Introduction

Pg 3, line 1: "SSA". This is the first time this is being mentioned in the main body, please write in full before introducing acronym in parentheses. Do same for all similar cases (e.g. MHC in line 29, CBC and SOP in Pg 4, line 26 etc).

line 2: "due  to  a  number" please replace due to with "because"

line 8: please write 3/4th in words. P.  falciparum. Please write generic name in full as this is the first time it is mentioned in the manuscript.

line 11: Anopheles, Plasmodium

Pg 4. Authors have a section titled materials and methods, and another section titled methods. I suggest the removal of "methods" and division of the materials and methods into subsections including study location and population, study design, inclusion and exclusion criteria, sample collection, data analysis etc.

line 20: "(Biftu  and ". Where does the parentheses end? ")"

line 22: remove MHC from parentheses.

Results

Pg 5, line 5: "A  total  of  206  study  participants  complete  the  study." please rephrase statement.

line 6: "Female  were  53.9  %  totally  and  higher  in  both  MHC  and  OH," please recast statement.

line 7: "were  37.8  %  with  majority  (26  %)" I suggest maintaining 1 decimal place all through manuscript. Please use 26.0% instead.

lines 5-16: This section needs to be revised by a native English speaker. The manner in which the data were presented is confusing. E.g. I think it would be better understood if authors wrote "A total of 37.8% of the study participants were rural dwellers while 62.2% were urban settlers. 26.0% of the rural dwellers had MHC, while 38.4% of the urban dwellers had OH". Please consider consulting a native English speaker for proof-reading and thorough revision of manuscript.

Also, I suggest presenting results for the entire study population first, then each of the subgroups (MHC and OH) to avoid confusion. Alternatively, results can be presented as done on lines 10-11 "Totally  54.8%  had  CD4  count  of  ≥500  cells/μl  (22.8%  MHC  vs.  32 % OH)"

line 10: "P≤0.02" please refer to my previous comment on this. Secondly, what confidence interval was used for this study? 95% or 99%? Statistical difference should be presented as P < 0.05, P > 0.05, P < 0.01 or P > 0.01. 0.01 is 99% confidence interval, not 0.001. If authors wish to present the P value, then it should be presented with an "=", e.g. p = 0.048 on line 27.

Authors wrote "Totally  54.8%  had  CD4  count  of  ≥500  cells/μl  (22.8%  MHC  vs.  32 % OH)" on lines 10-11, and then went ahead to write "Majority  of  OH  (64.07%)  had  CD4  count  of  ≥500  cells/μl  as  compared  with  MHC  (45.6%)" on line 13. This is confusing. Please revisit results.

lines 24-28: "There  was  no  significant  correlation..." where was this result presented? Please indicate table/figure.

Pg 6, lines 3-4: "Anemia  in  female  was  higher  than in  male  (61.3  %  vs.38.7  %)  among  anemic." please rephrase statement.

lines 8-28: please specify the tables/figures where these results were presented.

Pages 7-9: Authors did not sequentially discuss the results from the study. I suggest that authors discuss the results starting with socio-demographic and clinical characteristics, RBC indices in MHC and OH, correlation of CD4 count and RBC Indices in MHC and OH, etc.

There are also a lot of grammatical errors in this section. e.g."... finding of Nigeria" "Study of Ghana" (line 10) "Study  from ..." (lines 14 and 16) etc

Reviewer #2: The study was performed to find the correlation of RBC Indices and CD4 count in only HIV-infected (OH) and Malaria HIV co-infected (MHC) patients. RBC’s indices of MHC patients were significantly lower as compared to OH patients. On the other hand, a positive correlation was found between MHC and OH patients with respect to anemia and CD4 count. In MHC patients, CD4 count was positively correlated with RBC and Hgb. While in OH patients, CD4 count was positively correlated with MCV, MCH, and MCHC. While these observations in part confirm previous studies, the RBC indices in MHC and OH in Bench Maji Zone in SSA remain to be clarified. I request the authors to re-write the article as it is challenging to follow and incoherent. Please consider dividing the materials and methods with subheadings such as Study subjects, Determination CD4 counts, Statistics, etc. Consider stating a clear hypothesis in the introduction. Please consider proofreading for grammar as the document is full of mistakes.

6. PLOS authors have the option to publish the peer review history of their article (what does this mean?). If published, this will include your full peer review and any attached files.

Reviewer #1: **Yes: **Nneoma Confidence JeanStephanie Anyanwu

Reviewer #2: **Yes: **Himanshu Batra

---

## [Author Response · Author response to Decision Letter 0]

8 Dec 2021

Response to reviewers

Dear reviewers, First and above all I highly apologize for my inexperienced writing and lack of punctuality due to the human made (Instability and unstable politics) problems in the area where I am working and my personal problems although I have high enthusiasm to do my academic works. However, I got this few relative stability to finalize. Thanks for understanding me!!

Here under I attempt to address the comments provided by you.

1. I minimized the abbreviations and in place where I am supposed to use them I defined them clearly before using them 

2. I tried to minimize the inclusion bias among groups, particularly co-infected one; they were interviewed and assured that they were not sick of other chronic diseases. In this sense the co-infected group is in a similar state with the exception of acute malaria.

3. The study participants were duly acknowledged for their unreserved collaboration 

4. Before the collection of both laboratory data and data from interview I secured the consent of the participants both in written and oral form( the detail is available in major document) 

5. The detail sample size calculation procedure were followed

6. Yes the study have couple of limitations:-1) It did not assessed cause and effect as it is a crossectional study and 2) The CD4 count should be measured at the time of data collection 3) It would have been better if I had used PCR for diagnosis of malaria

7. The questionnaire tool (English version ) is attached at the last section of the document

8. The Ethical issue is secured from the Research Ethical Committee of Addis Ababa university and Bench Maji Health office 

9. I assign a number for each line and I tried to correct spelling errors 

10. Regarding the correlation of RBC indices and CD4 count the detail is located on table 4

11. All the spelling ,grammatical and abbreviations issue is secured

12. In material and method section I divided the subtitles with their respective heading 

13. Tables in this manuscript is attached at final section of the document

---

## [Decision Letter · Decision Letter 1]

31 Jan 2022

Effect of Malaria and HIV/AIDS co-infection on Red Blood Cell Indices and Its relation with the CD4 level of Patients on HAART in Bench Sheko Zone, Southwest Ethiopia.

PONE-D-21-03985R1

Dear Dr.Solomon Ejigu ,

We’re pleased to inform you that your manuscript has been judged scientifically suitable for publication and will be formally accepted for publication once it meets all outstanding technical requirements.

Kind regards,

Weijing He, M.D.

Academic Editor

PLOS ONE

Additional Editor Comments (optional):

Reviewers' comments:

Reviewer's Responses to Questions

**Comments to the Author**

1. If the authors have adequately addressed your comments raised in a previous round of review and you feel that this manuscript is now acceptable for publication, you may indicate that here to bypass the “Comments to the Author” section, enter your conflict of interest statement in the “Confidential to Editor” section, and submit your "Accept" recommendation.

Reviewer #1: All comments have been addressed

Reviewer #2: All comments have been addressed

2. Is the manuscript technically sound, and do the data support the conclusions?

Reviewer #1: Yes

Reviewer #2: Yes

3. Has the statistical analysis been performed appropriately and rigorously? 

Reviewer #1: (No Response)

Reviewer #2: Yes

4. Have the authors made all data underlying the findings in their manuscript fully available?

Reviewer #1: (No Response)

Reviewer #2: Yes

5. Is the manuscript presented in an intelligible fashion and written in standard English?

Reviewer #1: Yes

Reviewer #2: Yes

6. Review Comments to the Author

Reviewer #1: (No Response)

Reviewer #2: Thanks for addressing my comments during the unrest in your area. I sincerely appreciate it.

7. PLOS authors have the option to publish the peer review history of their article (what does this mean?). If published, this will include your full peer review and any attached files.

Reviewer #1: **Yes: **Nneoma Confidence J Anyanwu (Ph.D)

Reviewer #2: **Yes: **Himanshu Batra

---

## [Editor Report · Acceptance letter]

4 Feb 2022

PONE-D-21-03985R1 

Effect of Malaria and HIV/AIDS co-infection on Red Blood Cell Indices and Its relation with the CD4 level of Patients on HAART in Bench Sheko Zone, Southwest Ethiopia. 

Dear Dr. Ejigu:

I'm pleased to inform you that your manuscript has been deemed suitable for publication in PLOS ONE. Congratulations! Your manuscript is now with our production department. 

Kind regards, 

on behalf of

Dr. Weijing He 

Academic Editor

PLOS ONE